# Impact of Alloy Elements on the Adsorption and Dissociation of Gaseous Hydrogen on Surfaces of Ni–Cr–Mo Steel

Zhishan Mi [1], Xiuru Fan [2,*], Tong Li [1], Li Yang [1,*], Hang Su [1], Weidong Cai [1], Shuangquan Li [3] and Guoxin Zhang [3]

[1] Material Digital R&D Center, China Iron & Steel Research Institute Group, Beijing 100081, China; mizhishan@163.com (Z.M.); litongcisri@163.com (T.L.); hangsu@vip.sina.com (H.S.); gsgxwcwd@163.com (W.C.)

[2] College of Mechanical Engineering and Automation, Dalian Polytechnic University, Dalian 116034, China

[3] Sinopec Guangzhou Engineering Co., Ltd., Guangzhou 510600, China; lisq.lpec@sinopec.com (S.L.); zhanggx.lpec@sinopec.com (G.Z.)

* Correspondence: fanxiuru@dlpu.edu.cn (X.F.); yangli_drc@163.com (L.Y.)

**Abstract:** In this study, the effect of alloying elements on the adsorption and dissociation behaviors of hydrogen molecules on the bcc-Fe (001) surface has been investigated using first-principles calculations. $H_2$ molecules can easily dissociate on the hollow site, and the dissociated hydrogen atoms bond with the surrounding metal atoms. Doping Cr and Mo atoms on the surface would reduce the $H_2$ molecule adsorption energy, which promotes the $H_2$ molecule adsorption and dissociation. When only one or two Ni atoms doping on the surface, it improves the adsorption energies, which in turn can hinder the $H_2$ molecule adsorption and dissociation. However, three or four Ni atoms doping on the surface is beneficial to the $H_2$ molecule adsorption and dissociation. Thus, the nickel content in Ni–Cr–Mo steel should be reasonably controlled to improve the hydrogen embrittlement resistance of the steel.

**Keywords:** hydrogen adsorption energy; hydrogen embrittlement resistance; Ni–Cr–Mo steel; first-principles calculations





## 1. Introduction

As a clean, renewable, and zero-emission fuel, hydrogen energy has been wildly acknowledged as one of the most important energy sources in the coming decades [1,2]. However, the challenge of storing hydrogen has hindered its large-scale application due to its light molecular weight. High-pressure hydrogen storage and gas transportation pipelines are the most economically and practically viable solutions [3,4]. For this reason, high-strength low-alloy (HSLA) steels attract attention as the candidate materials for transport pipelines and high-pressure vessels due to their superior combination of strength, toughness, and weldability. However, high pressure hydrogen may lead to a risk of hydrogen-induced embrittlement (HE) for structural materials [5,6], and the deterioration of mechanical properties due to the HE phenomenon might directly threaten the safety of the transportation and storage of hydrogen [7–9]. Thus, the evaluation of HE susceptibility and the investigation of HE mechanism in HSLA steels have become a research hotspot in recent years.

As a typical type of HSLA steel, Ni–Cr–Mo steel has attracted much attention since it has been widely used in the petrochemical industry, having been both extensively studied and rapidly developed recently. The mechanisms for HE of HSLA steels have been proposed and controverted between several theories, including hydrogen-enhanced decohesion (HEDE) [10], hydrogen-enhanced local plasticity (HELP) [11] and adsorption-induced dislocation emission (AIDE) [12]. These theories suggest that under high-pressure hydrogen atmospheres, the molecular hydrogen dissociated to atomic hydrogen at the surface

after its adsorption. Then, the hydrogen atoms permeate and diffuse into the bulk metal materials, leading to the deterioration of the materials' mechanical properties. Regarding the interaction between dissolved H atoms with the material and its mechanism, as well as the diffusion of H atoms in bulk materials, a variety of research has been undertaken [13–16]. Due to the small size of hydrogen atoms, the adsorption and dissociation of gaseous hydrogen on the surfaces of steel are not able to be directly observed using current experimental methods. Little research has focused on the adsorption and dissociation of hydrogen molecules on steel surfaces.

Fortunately, with the development of computational materials science, first-principles calculations based on density functional theory (DFT) are an effective means for theoretical studies of materials' physical and chemical properties at the atomic scale [17–20]. Tateyama and Ohno [21] discovered that the existence of hydrogen promotes the formation of vacancy in body-centered cubic (bcc) Fe, and monovacancies with two H atom (V-H2) are the major complex in ambient conditions of hydrogen pressure. According to Hayward and Fu's research [22], hydrogen atoms prefer to occupy the tetrahedral site (T site) in the bulk structure, but when trapped at a vacancy, it prefers an octahedral site (O site). Additionally, a monovacancy in bcc-Fe has the capacity to trap up to five hydrogen atoms. In our previous research [23], we also investigated the interaction of common alloying elements with H atoms and the impact on HE susceptibility in HSLA steel. F. Bozso et al. [24] found that the adsorption energies of H on low indices of $\alpha$-Fe surface were similar, e.g., the adsorption energies on Fe (110), (100), and (111) surfaces measured to be 26, 24, and 21 kcal/mol, respectively. M. Li et al. [25] discussed the adsorption and dissociation of high-pressure hydrogen on Fe (100) and $Fe_2O_3$ (001) surfaces based on DFT calculation. However, the impact of alloying elements on the adsorption and dissociation of gaseous hydrogen on surfaces in Ni–Cr–Mo steel has not been discussed in detail so far.

In this study, density functional theory (DFT) calculations were performed to study the effect of alloying elements on the adsorption and dissociation behaviors of hydrogen molecules on a steel surface. The bcc-Fe (001) surface was selected as a research model. The Fe atom on the surface was substituted with Ni, Cr, or Mo atoms, which are common additional elements in Ni–Cr–Mo steels. The hydrogen adsorption energies on the clean and alloying atoms' doping surface were calculated. Additionally, the charge-transfer and density of states (DOS) analyses were explored in detail. Our calculated results provide theoretical guidance for designing HSLA steel with low hydrogen embrittlement susceptibility.

## 2. Computational Methods

All calculation results performed within the first-principles calculations based on DFT [26,27] were performed using the commercial Vienna Ab initio Software Package (VASP) code [28,29]. The electronic energy was obtained using PAW (projected augmented wave) potentials [30], and the Perdew–Burke–Ernzerhof (PBE) functional was used for core and valence electrons exchange, as well as correlation [31]. All energies were converged within a cutoff energy of 500 eV. All atoms were fully relaxed with the residual force less than 0.01 eV/Å on each atom, and the total energy was converged to $10^{-5}$ eV/atom for the electronic self-consistent field (SCF) calculations. Spin polarization was applied on all the Fe (001) surface and doping models. Convergence was determined using the tetrahedron method, implementing Blöchl-corrected smearing [32]. The Gaussian smearing width was set 0.05 eV for all the structures. The Brillouin zone (BZ) $5 \times 5 \times 1$ K-points reciprocal lattice matrix was generated using the Monkhorst–Pack method [33]. For density of states (DOS) calculations, a much denser K-points mesh ($7 \times 7 \times 1$) was used to obtain the accurate DOS of the surface model. A post stage vdW DFT-D3 method with Becke–Jonson damping was applied [34]. All the structural models and charge density diagrams were obtained using the visualization software VESTA ver. 3.5.8.

### 3. Results and Discussion

The adsorption and dissociation behaviors of $H_2$ molecules were first investigated on a clean bcc-Fe surface. The lattice constants of bcc-Fe unit cell a = b = c = 2.834 Å were from our previous calculations [23], which were consistent with the experimental values a = b = c = 2.86 Å [35]. The model used in the surface calculations is shown in Figure 1a. It was obtained after cleaving bulk iron with BCC crystal lattice at the surface plane with (001) Miller indices. Considering periodic boundary conditions and computer power, a 3 × 3 × 4 bcc-Fe supercell was used to keep the distance far enough between the doping defects. Thus, interaction between the periodic defects could be ignored. The 3 × 3 × 4 bcc-Fe supercell includes 72 Fe atoms with the lattice constants a = b = 8.599 Å. The vacuum layer was at least 20 Å above the surface. The distance between the nearest neighboring Fe atoms on the surface was optimized to 2.866 Å. Five top layers of the surface model shown were relaxed, whereas the three bottom layers from the model were kept fixed.

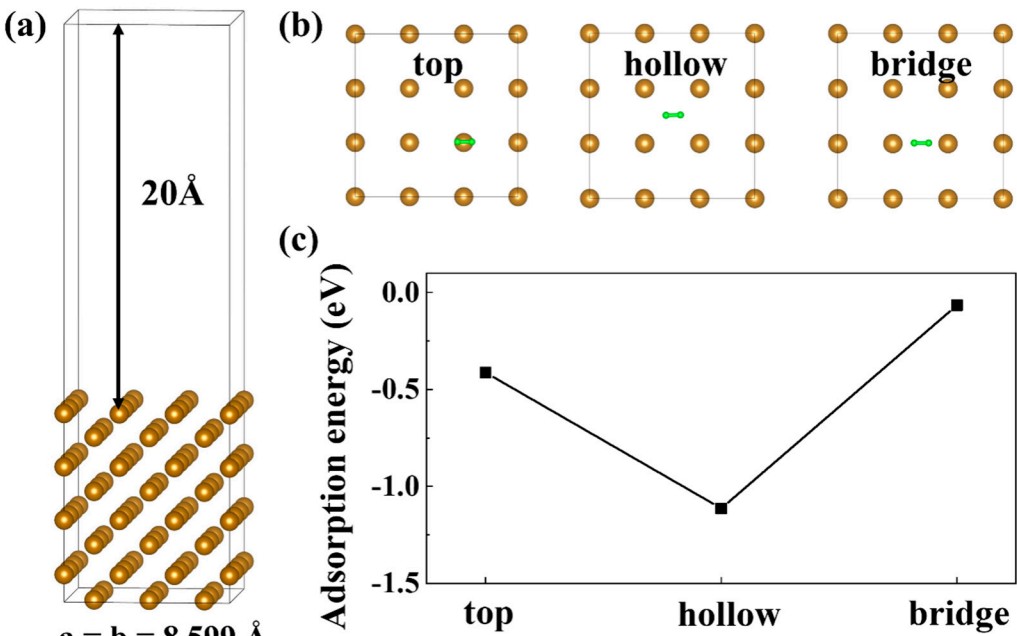

**Figure 1.** (**a**) BCC-Fe (001) surface model; (**b**) Three different starting configurations for hydrogen adsorption on the bcc-Fe (001) surface; (**c**) Hydrogen adsorption energies of three configurations.

The adsorption energies of $H_2$ molecules on a clean bcc-Fe (001) surface were calculated first. The adsorption energy $E_{ads}$ was defined as:

$$E_{ads} = E_{(surf\ with\ H_2)} - E_{(surf)} - E_{(H_2)} \tag{1}$$

where $E_{(surf\ with\ H_2)}$ is the total energy of the surface model with a single $H_2$ molecule after the structural relaxation; $E_{(surf)}$ is the obtained total energy of the constructed surface model without any molecule adsorption; and $E_{(H_2)}$ is referenced to the calculated total energy of a single free $H_2$ molecule. Hence, the lower adsorption energy indicates the surface model was more favorable to adsorbing $H_2$ molecules.

The $H_2$ adsorption energies of different locations were calculated, and the most stable one was chosen for further calculation and discussion. Generally, spontaneous chemisorption was observed to occur when the molecule was placed less than 2.0 Å above the topmost surface atomic plane. In our studies, the $H_2$ molecule was placed 1.6 Å over the bcc-Fe (001) surface and fully relaxed. Figure 1b displays the three different starting configurations for hydrogen adsorption on the bcc-Fe (001) surface, including the top site, hollow site, and bridge site.

The adsorption energies of three configurations in Figure 1c were calculated using Formula (1). The calculated $E_{ads}$ on the hollow site was the lowest, about −1.113 eV. The adsorption of hydrogen molecules was strongest on the hollow site. A negative value for the adsorption energy represented an energy release when the $H_2$ molecule is adsorbed on the hollow site of the bcc-Fe (001) surface.

The three configurations after structural relaxation are shown in Figure 2a–c. The $H_2$ molecule initially placed at the hollow site dissociated into two H atoms with an H–H distance of 2.982 Å, suggesting a completely broken H–H bond (0.751 Å). These H atoms were stabilized at two octahedra sites above the bcc-Fe (001) surface with a Fe–H bond length of 1.700 Å. Previous studies [25] have also shown that the $H_2$ molecule adsorbed on the Fe (100) surface dissociates into two hydrogen atoms with an adsorption energy of −0.89 eV, which demonstrated the reliability of our calculation results. In contrast, the $H_2$ molecules initially placed at the top site and bridge site were not dissociated into two H atoms, with varying H–H bond lengths of 0.878 Å and 0.757 Å, respectively. Hence, the initial configurations of the top and bridge sites for $H_2$ molecule adsorption would have been unstable.

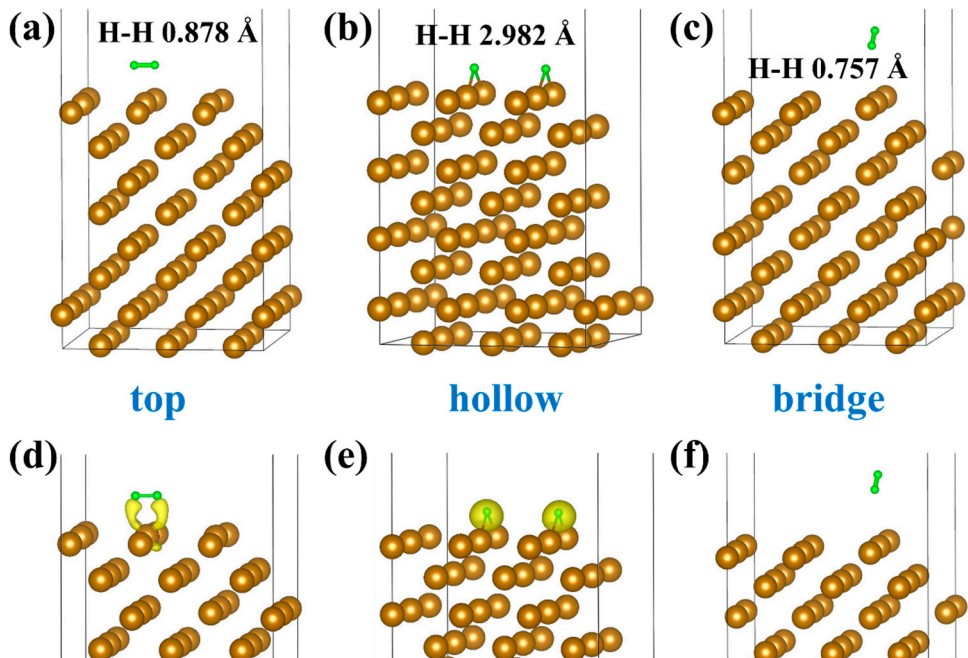

**Figure 2.** The adsorption and dissociation configurations of a $H_2$ molecule at (**a**) top, (**b**) hollow, and (**c**) bridge site on the Fe surface after structural relaxation. The differential charge density of $H_2$ on the (**d**) top, (**e**) hollow and (**f**) bridge sites after the structural relaxation.

Charge transfer analysis was performed utilizing DFT calculation to understand the bonding mechanism of the $H_2$ molecule adsorbed at different sites on Fe surface. First, the valence electron distributions of $H_2$ on the top, hollow, and bridge sites after adsorption were analyzed using Bader Charge Analysis. The valence electron numbers for two H atoms on the top, hollow, and bridge sites were (1.074$e$, 1.062$e$), (1.369$e$, 1.369$e$), and (1.033$e$, 0.997$e$), respectively. As the first element in the periodic table, the hydrogen atom has only one valence electron outside its nucleus. Our calculated results indicated that hydrogen atoms on hollow site could gain more electrons and have strong interaction with Fe atoms on the surface.

In addition, the differential charge density for $H_2$ on the top, hollow, and bridge sites was defined as:

$$\rho = \rho_{(model\ with\ H_2)} - \rho_{(model)} - \rho_{(H_2)} \qquad (2)$$

where $\rho_{(model\ with\ H_2)}$ was the total charge density of the surface model adsorbed with a $H_2$ molecule, and $\rho_{(model)}$ and $\rho_{(H_2)}$ were the charge density of the surface model and $H_2$ molecule, respectively. The differential charge density of $H_2$ on the top, hollow, and bridge sites after the structural relaxation have been illustrated in Figure 2d–f. The iso-surface of charge density was set to 0.008 e/$Å^3$. The yellow areas represent valence electrons increment of the H atom or $H_2$ molecule. Obviously, the two H atoms on the hollow site gained more valence electrons than the $H_2$ molecule on the top site, and resulted in a more restrained bonding of H to Fe atoms. For the surface model of the $H_2$ molecule on the bridge site, there was no charge transfer interaction between the $H_2$ molecule and the Fe atoms on the surface.

Alloying elements for HSLA steel are added to steel to gain excellent performance, including mechanical properties and corrosion resistance characteristics. These alloying elements may exist on the surface and interior of the steel. To further understand the role of alloying elements (Ni, Cr, and Mo) for Ni–Cr–Mo steel in hydrogen resistance characteristics, the hydrogen adsorption of alloying elements doped Fe surface model were studied.

Doping effects of Ni, Cr, and Mo atoms in the bcc-Fe (001) surface model were the first to be studied. One Fe atom in the Fe surface was substituted with one Ni, Cr, and Mo atom, in which the doping concentration was 1.39 at% (atomic percentage). One alloying atom replaced the Fe atom in different sites from the top layer to the center of the surface model to find the most energetic doping site, which was denoted by layer 1 to 5 of the surface model as presented in Figure 3a. To confirm the structural stability of the Ni, Cr and Mo substitutions in the Fe surface model, the formation energies were calculated according to:

$$E_f = E_{(doped\ surf)} - E_{(surf)} + \mu_{(Fe)} - \mu_{(X)}. \tag{3}$$

where $E_{(doped\ surf)}$ and $E_{(surf)}$ were the total energies derived from the doped and clean Fe surface models after structural relaxation, respectively. $\mu_{(Fe)}$ and $\mu_{(X)}$ represent the chemical potentials of Fe and alloying atoms (Ni, Cr and Mo), respectively, which were calculated based on the cubic metals. Hence, a lower formation energy indicated the alloying atom was more stable in this surface model.

Figure 3b plots the formation energies of the doped Fe surface with different doping sites using Formula (3). It could be found that doping Ni in all the studied sites was stable, with negative formation energies, and the doping surface model with the outermost Fe atom (layer 1) replaced with Ni had the lowest formation energy of −0.339 eV. When one Cr or Mo atom replaced the Fe atom in the location (layer 5) inner the model, the formation energy was the lowest, corresponding to the formation energy of −0.308 eV for the Cr doping model and −0.222 eV for the Mo doping model. These results suggest that the Ni atoms preferred to diffuse to the surface of the Fe surface model, while the Cr and Mo atoms were easy to diffuse inside the model. Therefore, in order to investigate the effect of alloying elements on $H_2$ molecule adsorption in more detail, the $H_2$ adsorption energies of Ni, Cr, and Mo doping on the surface and inside were calculated and compared in the following study.

For alloying atoms doping surface model, the different $H_2$ adsorption sites were also calculated, including the top, hollow, and bridge sites. By comparison to the adsorption energies, the hollow site was most stable for all alloying atoms doping models. Figure 3c,d display the $H_2$ adsorption energies on the hollow site of Ni, Cr, and Mo doping both inside and on the surface, calculated using Formula (1). In the case of Ni, Cr, and Mo doping inside the Fe surface model, the adsorption energies were almost consistent with the case without alloying doping, while the $H_2$ molecules dissociated into two hydrogen atoms on the surface due to the low adsorption energies. As a Ni, Cr, or Mo atom doping inside, the atomic site occupation of Fe atoms on the surface almost did not change. Thus, the doping of internal alloying elements had no effect on the hydrogen adsorption and dissociation on the surface.

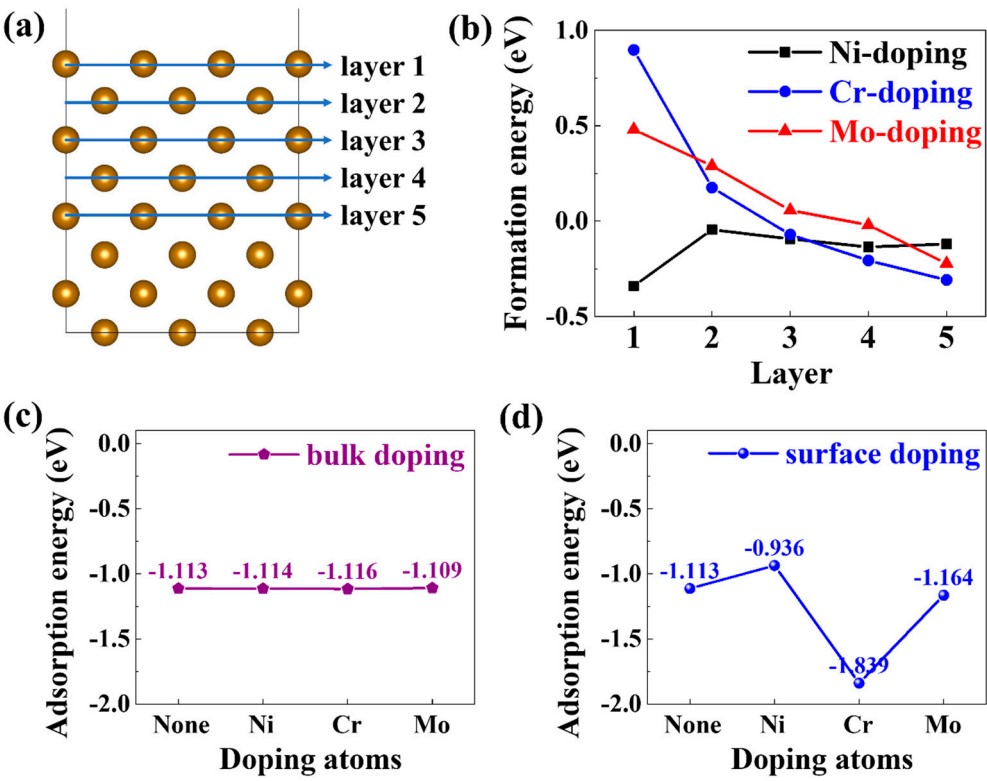

**Figure 3.** (**a**) Different doping sites of alloying atom in the Fe surface; (**b**) Formation energies of the alloying atoms in different doping sites. $H_2$ adsorption energies on the hollow site of clean, Ni, Cr, and Mo atoms doping (**c**) inside and (**d**) on the surface.

From Figure 3d, it can be seen that the alloying elements doping on the surface have a significant effect on $H_2$ molecule adsorption energies. For the case of Ni doping on the surface, the adsorption energy was $-0.936$ eV, which is 0.177 eV higher than that of the surface without doping. The increasing adsorption energy demonstrated that Ni atom doping hinders $H_2$ molecule adsorption on the surface. However, in the case of the Cr and Mo doping surfaces, the adsorption energies reduced to $-1.839$ eV and $-1.164$ eV, suggesting that Cr and Mo atoms doping promotes the $H_2$ molecule adsorption on the surfaces. The optimized geometric structures and differential charge density diagrams of $H_2$ molecule adsorption on the hollow site of Ni, Cr, and Mo atoms doping on the surface have been presented in Figure 4a–c. It can be seen that the $H_2$ molecules initially placed at the hollow site were still dissociated into two hydrogen atoms on all alloying atom doping surface. These dissociated hydrogen atoms become stabilized at octahedra sites above the surface and bond with the surrounding Fe atoms and alloying atoms. For Ni atom doping surfaces, the Ni–H bond length was 1.611 Å, which is shorter than the Fe–H bond length (1.700 Å). For Cr and Mo atom doping surfaces, the Cr–H and Mo–H bond lengths were 1.801 Å and 1.877 Å, respectively. Although the Ni–H bond length was shorter, the $H_2$ molecule had higher adsorption energy on the Ni atom doping surface, which indicated that the alloying element had more of an effect on the $H_2$ molecule adsorption and dissociation behavior. Through the charge transfer analysis in Figure 4d, the hydrogen atom obtained fewer electrons due to Ni atom doping, while the hydrogen atoms on the Cr and Mo atom doping surface obtained more electrons. According to the Periodic Table of Elements, the arrangements of extranuclear electrons are $[Ar]3d^6\,4s^2$ for Fe, $[Ar]3d^8\,4s^2$ for Ni, $[Ar]3d^5\,4s^1$ for Cr, and $[Kr]4d^4\,4s^1$ for Mo. The Ni atom has more electrons outside the nucleus and was less attractive to the hydrogen atom. From our calculated results, the alloying elements doping on the surface of steel can influence the $H_2$ molecule adsorption and dissociation behavior. It is worth noting that Ni atoms were more likely to exist on the surface, while Cr and Mo atoms tended to exist in the structure according to the stability calculation of

the alloying element doping model. Thus, the effect of Ni atoms doping on $H_2$ molecule adsorption on the surface needs further attention.

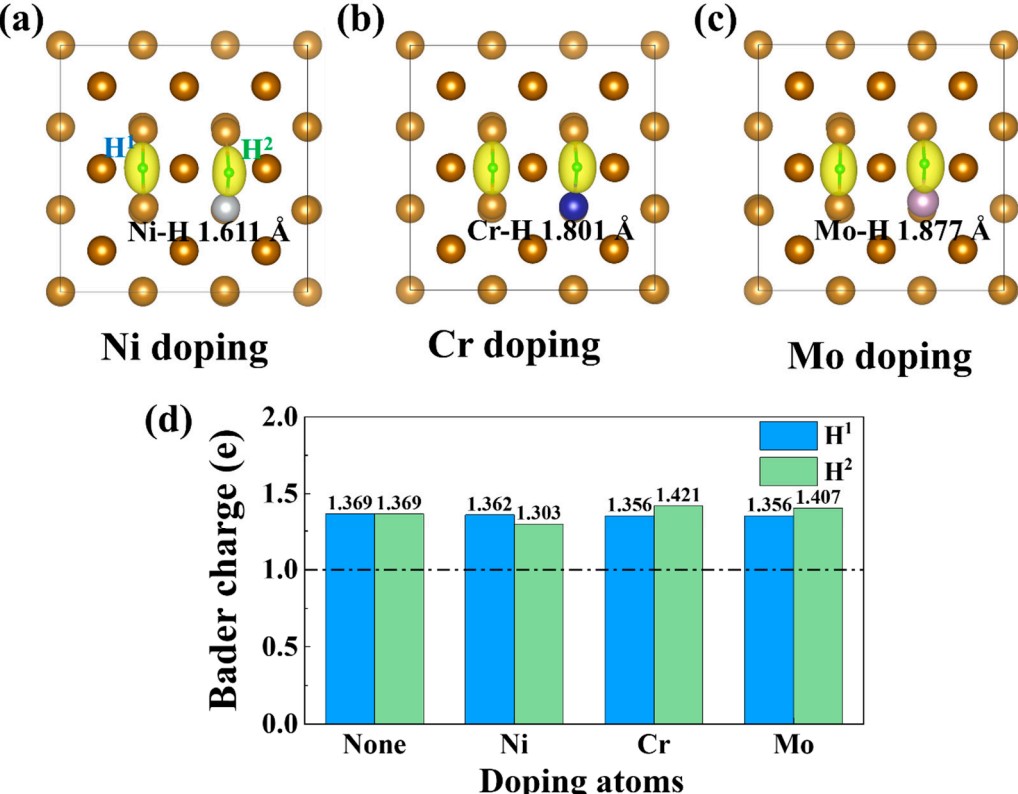

**Figure 4.** The optimized geometric structures and differential charge density diagrams of $H_2$ molecule adsorption on the hollow site of (**a**) Ni atom, (**b**) Cr atom, and (**c**) Mo atom doping on the surface; (**d**) Bader charge analysis of two H atoms on the clean and doping surface.

To investigate the effect of doping concentration on $H_2$ molecule adsorption, multiple Ni atoms doping on the surface were calculated. Due to the two dissociated hydrogen atoms, four Fe–H bonds formed with the surrounding Fe atoms, and a further two to four Ni atoms doping on the surface were studied. The Ni atoms doping concentration ranged from 2.78 at% to 5.56 at%. These Ni atoms were stabilized at the bcc lattice position after structural relaxation, only a slight deviation from the lattice position. In the following, the $H_2$ molecule was placed on the surface with different Ni atoms doping and fully relaxed.

Figure 5a–c show the optimized structures of $H_2$ molecule adsorption on the hollow site with different Ni atoms doping. It can be seen from the atomic structure diagram that the $H_2$ molecule dissociated into two hydrogen atoms and bonded to the surrounding Ni or Fe atoms in all cases. The adsorption energies of hydrogen molecules on multiple Ni atoms doping on the surface were further calculated (c.f. Figure 5d). The adsorption energy gradually decreased with the increase in doped Ni atoms, indicating that more doped Ni atoms would promote the adsorption and dissociation behavior of hydrogen molecules. However, the $H_2$ adsorption energy of two Ni atoms doping on the surface was higher than that none–Ni atom doping, while the adsorption energies of three and four nickel atoms doping were lower than that none–Ni atom doping. Such calculated results indicated that doping a small amount of Ni in Fe can hinder the $H_2$ molecule adsorption, while more Ni atoms doping might lead to the opposite results. Previous experimental studies [36,37] have shown that nickel content has an effect on the hydrogen resistance of steel materials.

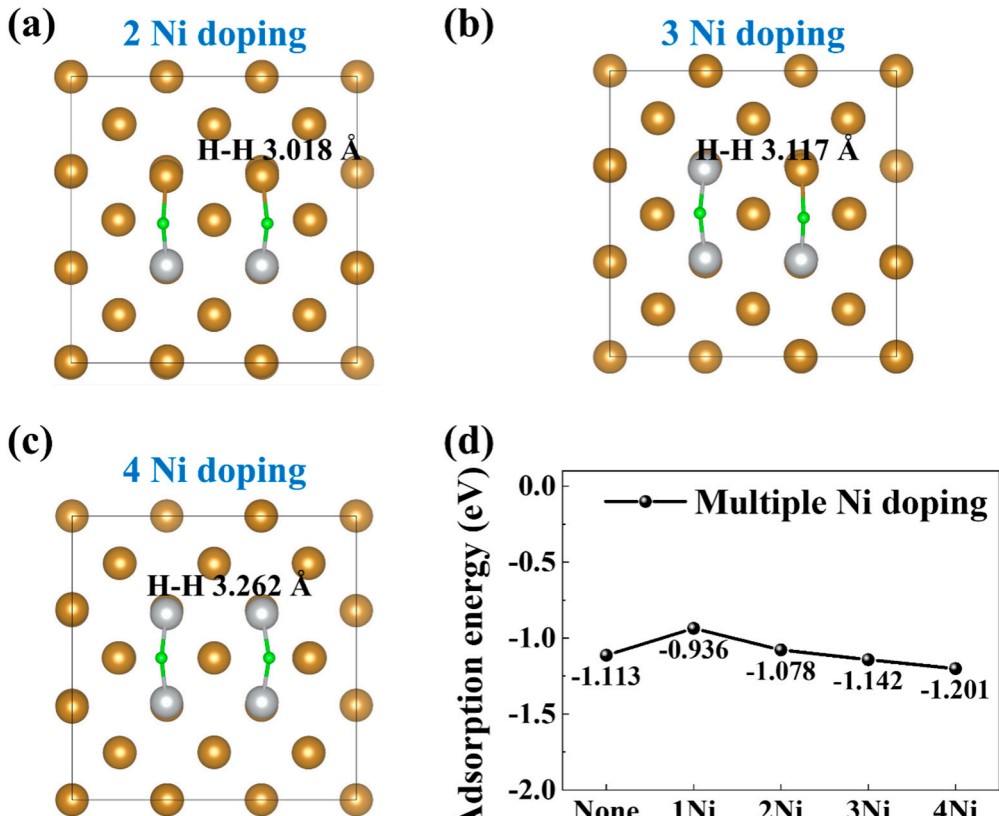

**Figure 5.** The optimized geometric structures of $H_2$ molecule adsorption on the Fe (001) surface with (**a**) two Ni atoms, (**b**) three Ni atoms, and (**c**) four Ni atoms doping; (**d**) the adsorption energies of $H_2$ molecules on multiple Ni atoms doping on the surface.

It is important to understand the influence mechanism of the nickel content on hydrogen resistance from the electronic structure. The density of states (DOS) patterns can provide the information of bonds between various orbitals, which were then employed to study the bonding mechanism of H and metal atoms. The DOS of $H_2$ molecule adsorption on bcc-Fe (001) surface with different Ni atoms doping was conducted as shown in Figure 6. The Fermi level was set to 0 eV. The black line in Figure 6 corresponds to the total density of states (TDOS) of the model, and the blue, green, and red lines represent the partial density of states (PDOS) of the Fe, H, and Ni, respectively. A positive value of DOS represents the spin-up electronic states and a negative value of DOS represents the spin-down electronic states. It was shown that the major DOS of the H atoms were in the energy range of $-4$ to $-8$ eV, and the major DOS of the Ni atoms were in the energy range of 0 to $-5$ eV below the Fermi level. The TDOS above the Fermi level were dominated by the Fe-3d states. It is worth noting that the Ni-3d states were far away from the Fermi level with the Ni atoms increasing, which led to more Ni-3d states and H-1s states in the same energy level. When the DOS of the two elements were at the same energy level, it showed that their charge electrons had a strong interaction. Our results showed that more Ni atoms doping would enhance the interaction between Ni and H atoms, which is beneficial to hydrogen molecules adsorbing and dissociating on the surface. Thus, the hydrogen embrittlement resistance of steel can be further improved by adjusting the proportion and position of alloying elements.

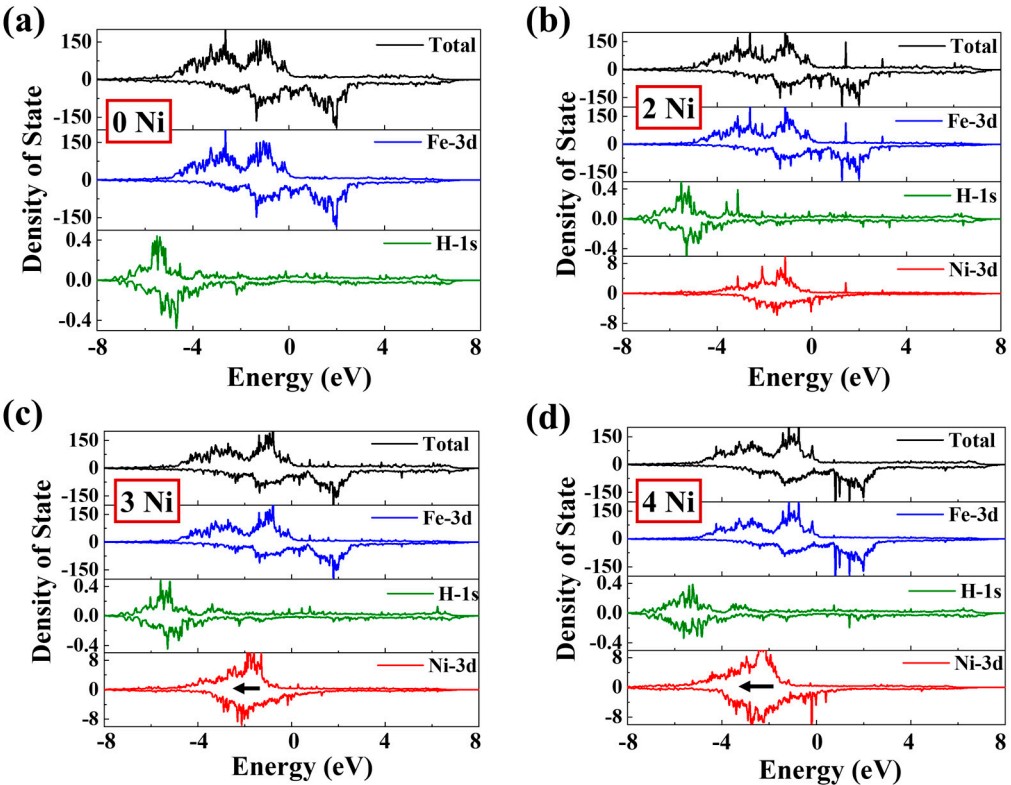

**Figure 6.** TDOS and PDOS of $H_2$ molecule adsorption on bcc-Fe (001) surface with different Ni atoms doping; (**a**) No Ni atoms doping, (**b**) two Ni atoms doping, (**c**) three Ni atoms doping, (**d**) four Ni atoms doping.

## 4. Conclusions

In summary, the adsorption and dissociation behaviors of $H_2$ molecule on the clean bcc-Fe (001) surface and alloying atoms doping surface were investigated through first-principles calculations. The following conclusions, along with some insights for controlling HE in steels, were made:

(1) The starting adsorption sites were important for the dissociation of $H_2$ molecule on the bcc-Fe (001) surface. $H_2$ molecules could easily dissociate and form Fe–H bonds on the hollow site of clean surface with an adsorption energy of $-1.113$ eV.

(2) Alloying elements doping on the surface obviously affected the $H_2$ adsorption energies while doping in the inner layer had almost no effect. Ni atoms were more likely to exist on the surface, and Cr or Mo atoms tended to exist within the surface model.

(3) Cr and Mo atoms doping on the bcc-Fe (001) surface reduced the $H_2$ molecule adsorption energy by 0.726 eV (Cr) and 0.051 eV (Mo), respectively. The results suggested the Cr and Mo atoms on the Fe surface promoted the $H_2$ molecule adsorption and dissociation, increasing the possibility of hydrogen permeation.

(4) One or two Ni atoms doping on the surface improved the adsorption energies by 0.177 eV and 0.035 eV, which could hinder the $H_2$ molecule adsorption and dissociation. However, three or four Ni atoms doping on the surface was beneficial to the $H_2$ molecule adsorption and dissociation.

Therefore, it is possible to effectively enhance the hydrogen embrittlement resistance by controlling the nickel content in the steel. Our calculation results provide guidance for the design of advanced steel materials with low hydrogen embrittlement susceptibility steels in industry.

**Author Contributions:** Conceptualization, Z.M., X.F., L.Y. and H.S.; methodology, Z.M., X.F. and T.L.; investigation, Z.M., X.F., H.S. and W.C.; data curation, Z.M., T.L., S.L. and G.Z.; writing—original draft preparation, Z.M. and T.L.; writing—review and editing, Z.M., L.Y. and H.S.; visualization, Z.M., X.F. and W.C.; project administration, Z.M. and L.Y. All authors have read and agreed to the published version of the manuscript.

**Funding:** This research was funded by the National Key Research and Development Program of China (Grant No. 2022YFB4003001).

**Data Availability Statement:** The data presented in this study are available upon request from the corresponding author.

**Conflicts of Interest:** Authors Z.M., T.L., L.Y., H.S. and W.C. were employed by the company China Iron & Steel Research Institute Group, S.L. and G.Z. were employed by the company Sinopec Guangzhou Engineering Co., Ltd. The remaining authors declare that the research was conducted in the absence of any commercial or financial relationships that could be construed as a potential conflict of interest.

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
