# Peer review of "Impact of Alloy Elements on the Adsorption and Dissociation of Gaseous Hydrogen on Surfaces of Ni–Cr–Mo Steel"

_processes, doi:10.3390/pr11113241_

Round 1

Reviewer 1 Report

Comments and Suggestions for Authors

The article titled "Impact of alloy elements on the adsorption and dissociation of 2 gaseous hydrogen on surfaces of Ni-Cr-Mo steel" is generally well written.

The following issues have been identified:

1. Review lines 34/35, 67 etc. and the general manuscript for grammar.

2. Methods are not well detailed. More information on input parameters and baseline conditions are needed. In its current state, it will be difficult for another researcher to replicate.

3.Improve on the discussion. For example, 1 or 2 Ni reduces increases energy but 3 or 4 reduces the energy level. Why might this be? Try and offer explanation for observed phenomenon.

4. Improve on the reference list. More than 50% of the references are over 10 years old.

Comments on the Quality of English Language

Some proofreading needed.

Author Response

Comments 1: Review lines 34/35, 67 etc. and the general manuscript for grammar.

Response 1: Thank you very much for taking the time to review this manuscript. Revised as required.

Lines 34/35: Thus, the evaluation of HE susceptibility and the investigation of HE mechanism in HSLA steels have become a research hotspot in recent years.

Line 67: However, the impact of alloying elements on the adsorption and dissociation of gaseous hydrogen on surfaces in Ni-Cr-Mo steel has not been discussed in detail so far.

Comments 2: Methods are not well detailed. More information on input parameters and baseline conditions are needed. In its current state, it will be difficult for another researcher to replicate.

Response 2: Revised as required. The description of input parameters is added in Computational Methods.

The Gaussian smearing width was set 0.05 eV for all the structures.

For density of states (DOS) calculations, a much denser K-points mesh (7 × 7 × 1) was used to obtain the accurate DOS of the surface model.

Comments 3: Improve on the discussion. For example, 1 or 2 Ni reduces increases energy but 3 or 4 reduces the energy level. Why might this be? Try and offer explanation for observed phenomenon.

Response 3: Revised as required. In the lines 258 to 277 of Results and Discussions, the influence mechanism of the nickel content on hydrogen resistance is explained by the density of states (DOS) diagram analysis. And a more detailed discussion is added.

It is important to understand the influence mechanism of the nickel content on hydrogen resistance from the electronic structure. The density of states (DOS) patterns can provide the information of bonds between various orbitals, which will be employed to study the bonding mechanism of H and metal atoms. The DOS of H2 molecule adsorption on bcc-Fe (001) surface with different Ni atoms doping was conducted as shown in Figure 6. The Fermi level is set to 0 eV. The black line in Figure 6 is corresponding to the total density of states (TDOS) of the model. And the blue, green and red lines represent the partial density of states (PDOS) of the Fe, H, and Ni, respectively. The positive value of DOS represents the spin-up electronic states and the negative value of DOS represents the spin-down electronic states. It is shown that the major DOS of the H atoms is in the energy range of - 4 to - 8 eV, and the major DOS of the Ni atoms is in the energy range of 0 to - 5 eV below the Fermi level. The TDOS above the Fermi level is dominated by the Fe-3d states. It is worth noting that the Ni-3d states are far away from the Fermi level with the Ni atoms increasing, which leads to more Ni-3d states and H-1s states in the same energy level. When the DOS of the two elements are at the same energy level, it shows that their charge electrons have a strong interaction. Our results show that more Ni atoms doping would enhance the interaction between Ni and H atoms, and is beneficial to hydrogen molecules adsorption and dissociation on the surface with more Ni doping. Thus, the hydrogen embrittlement resistance of steel can be further improved by adjusting the proportion and position of alloying elements.

Comments 4: Improve on the reference list. More than 50% of the references are over 10 years old.

Response 4: Revised as required. Some references in recent ten years are added.

7. Adasooriya, N. D.; Tucho, W. M.; Holm, E.; Arthun, T.; Hansen, V.; Solheim, K. G.; Hemmingsen, T., Effect of hydrogen on mechanical properties and fracture of martensitic carbon steel under quenched and tempered conditions. Materials Science and Engineering: A 2021, 803, 140495.

8. Ghosh, G.; Rostron, P.; Garg, R.; Panday, A., Hydrogen induced cracking of pipeline and pressure vessel steels: A review. Engineering Fracture Mechanics 2018, 199, 609-618.

11. Martin, M. L.; Dadfarnia, M.; Nagao, A.; Wang, S.; Sofronis, P., Enumeration of the hydrogen-enhanced localized plasticity mechanism for hydrogen embrittlement in structural materials. Acta Materialia 2019, 165, 734-750.

Reviewer 2 Report

Comments and Suggestions for Authors

The manuscript is devoted to the study of  the effect of alloying elements on the adsorption and dissociation behaviors of hydrogen molecules on steel surface. The manuscript is of great theoretical and practical interest.  The authors showed the possibility of effectively enhancing the hydrogen embrittlement resistance by controlling the nickel content in the steel. It provides guidance for the design of advanced steel materials with low hydrogen embrittlement susceptibility steels in industry.

The manuscript may be published as submitted.

Author Response

Thank you very much for taking the time to review this manuscript. We agree with this comment.

Reviewer 3 Report

Comments and Suggestions for Authors

The paper is interesting, and the topics discussed are particularly important in view of the current development of hydrogen technologies in transport and energy.

In my opinion, however, it requires some additions in the description of methods, research results and conclusions.

The actual structure of steel is not uniform in cross-section. There is the so-called a surface layer significantly different in structure and properties from the bulk material. Taking into account this layer and its heterogeneity in the models used is difficult and often impossible. The authors also presented an idealized model of the steel structure, where the only disturbances are atoms of doped elements and molecules and, after dissociation, hydrogen atoms. Resistance to hydrogen embrittlement is a bulk property of the material which, among other things, depends on the diffusion of hydrogen molecules and atoms. In my opinion, these aspects were not highlighted in the article. I suggest that the authors indicate this in the section regarding the research method, when discussing the research results, and especially in the conclusions. I believe that the reader should be clearly informed about the need to verify the presented conclusions

Author Response

Thank you very much for taking the time to review this manuscript. We agree with this comment. Currently, it is difficult to directly study the surface models of complex steel materials using first principles calculations. Therefore, in our study, we utilized an ideal structure as described in this paper, where the effects of complex surface structures were approximated by doping with different alloying elements. Our team has been engaged in the design of hydrogen-resistant high-strength steel and the investigation of hydrogen embrittlement mechanisms. In our previous work, we have studied the dissolution and diffusion behaviors of hydrogen atoms in BCC Fe structures and grain boundaries [1]. In our upcoming work, we plan to continue studying the inward permeation and diffusion behavior of hydrogen atoms on the surface of steel materials, aiming to provide a clearer understanding of the hydrogen embrittlement mechanisms.

[1] Fan, X.; Mi, Z.; Yang, L.; Su, H. Application of DFT Simulation to the Investigation of Hydrogen Embrittlement Mechanism and Design of High Strength Low Alloy Steel. Materials 2023, 16, 152. https://doi.org/10.3390/ma16010152.

Reviewer 4 Report

Comments and Suggestions for Authors

In this paper, the effect of alloying elements on the adsorption and dissociation behaviors
of hydrogen molecules on the bcc-Fe (001) surface was investigated using first-principles calculations. It was found that H2 molecule can easily dissociate on the hollow site. The dissociated hydrogen atoms bond with the surrounding metal atoms. Doping Cr and Mo atoms on the surface reduced the H2 molecule adsorption energy, which promotes the H2 molecule adsorption and dissociation. It was also shown that only one or two Ni atoms doping on the surface improves the adsorption energies, which can hinder the H2 molecule adsorption and dissociation. However, three and four Ni atoms doping on the surface are beneficial to the H2 molecule adsorption and dissociation. Recommendations were made to control the nickel content in Ni-Cr-Mo steel to improve the hydrogen embrittlement resistance of the steel.

 The paper could be considered for publication in the journal after the following major revisions:

1-Check the English of the paper.

2--use third person. Avoid using phrases like “We found” etc.

3-A better combination of keywords should be employed, especially avoid using failure analysis.

4-Define in the abstract what parameters were investigated, before briefly mentioning the results of such tests. Like mention that doping was done with how many atoms of what and emphasis on the experimental conditions in terms of process parameters.

5-Introduciton should be strengthened. To modify this section the following documents can be consulted:

-(2022). Water jet impact damage mechanism and dynamic penetration energy absorption of 2A12 aluminum alloy. Vacuum, 206, 111532. doi: https://doi.org/10.1016/j.vacuum.2022.111532

-(2019). A nonlinear attment-detachment model with adsorption hysteresis for suspension-colloidal transport in porous media. Journal of Hydrology, 578, 124080. doi: https://doi.org/10.1016/j.jhydrol.2019.124080

6-Better define the rationale of the work at the end of the introduction by linking the literature with the undertaken research.

7-it should be “Materials and Computational method”. Introduce the materials first. And make them as two subsections.

8-reference the formula that is used.

9-figure caption 2 is long and tedious.

10-figure 4 shows that Bader charge are rather small by doping. It is even negative for Ni addition. Can the authors elaborate more on this?

11-figure 6 is a bit messy. Make them more visible.

12-consult the following references in the discussion section:

-(2020). Evolution of crystallographic orientation, precipitation, phase transformation and mechanical properties realized by enhancing deposition current for dual-wire arc additive manufactured Ni-rich NiTi alloy. Additive Manufacturing, 34, 101240. doi: https://doi.org/10.1016/j.addma.2020.101240

-(2020). Hydrogen embrittlement behavior of SUS301L-MT stainless steel laser-arc hybrid welded joint localized zones. Corrosion Science, 164, 108337. doi: https://doi.org/10.1016/j.corsci.2019.108337

13-have the authors considered the atomic size difference of different doping? Or only bond energy was considered.

14-conclusions are long and tedious. Make them concise and shorter. It should reflect the results that were obtained, not known facts in the literature.

Comments on the Quality of English Language

1-Check the English of the paper.

2--use third person. Avoid using phrases like “We found” etc. 

Author Response

Comments 1: Check the English of the paper.

Response 1: Thank you very much for taking the time to review this manuscript. Checked and corrected as required.

Comments 2: use third person. Avoid using phrases like “We found” etc.

Response 2: Checked and corrected as required.

Comments 3: A better combination of keywords should be employed, especially avoid using failure analysis.

Response 3: Revised as required.

Comments 4: Define in the abstract what parameters were investigated, before briefly mentioning the results of such tests. Like mention that doping was done with how many atoms of what and emphasis on the experimental conditions in terms of process parameters.

Response 4: At present, it is difficult to observe the phenomenon at the atomic level by the experimental methods. We have not yet found the effect of the number of alloying elements on hydrogen adsorption in the experiment. First-principles calculation method is mainly to explain the interaction mechanism between different elements at the atomic level. It can provide a new perspective for researchers to understand hydrogen embrittlement.

Comments 5: Introduction should be strengthened. To modify this section the following documents can be consulted:

-(2022). Water jet impact damage mechanism and dynamic penetration energy absorption of 2A12 aluminum alloy. Vacuum, 206, 111532. doi: https://doi.org/10.1016/j.vacuum.2022.111532

-(2019). A nonlinear attment-detachment model with adsorption hysteresis for suspension-colloidal transport in porous media. Journal of Hydrology, 578, 124080. doi: https://doi.org/10.1016/j.jhydrol.2019.124080

Response 5: Revised as required.

Comments 6: Better define the rationale of the work at the end of the introduction by linking the literature with the undertaken research.

Response 6: Revised as required.

Comments 7: it should be “Materials and Computational method”. Introduce the materials first. And make them as two subsections.

Response 7: Our research object is bcc-Fe (001) surface, which has been introduced in the results and discussions.

Comments 8: reference the formula that is used.

Response 8: Revised as required.

Comments 9: figure caption 2 is long and tedious.

Response 9: Figure 2 shows the three configurations after structural relaxation and differential charge density of H2 on the top, hollow and bridge sites.

Comments 10: figure 4 shows that Bader charge are rather small by doping. It is even negative for Ni addition. Can the authors elaborate more on this?

Response 10: The arrangements of extranuclear electrons are [Ar]3d6 4s2 for Fe, [Ar]3d8 4s2 for Ni, [Ar]3d5 4s1 for Cr, and [Kr]4d4 4s1 for Mo. The Ni atom has more electrons outside the nucleus and is less attractive to the hydrogen atom.

Comments 11: figure 6 is a bit messy. Make them more visible.

Response 11: Revised as required. The density of states diagram is difficult to be clear.

Comments 12: consult the following references in the discussion section:

-(2020). Evolution of crystallographic orientation, precipitation, phase transformation and mechanical properties realized by enhancing deposition current for dual-wire arc additive manufactured Ni-rich NiTi alloy. Additive Manufacturing, 34, 101240. doi: https://doi.org/10.1016/j.addma.2020.101240

-(2020). Hydrogen embrittlement behavior of SUS301L-MT stainless steel laser-arc hybrid welded joint localized zones. Corrosion Science, 164, 108337. doi: https://doi.org/10.1016/j.corsci.2019.108337

Response 12: Revised as required.

Comments 13: have the authors considered the atomic size difference of different doping? Or only bond energy was considered.

Response 13: For the influence of different elements, we only do the single atom doping. The atomic size difference has little effect.

Comments 14: conclusions are long and tedious. Make them concise and shorter. It should reflect the results that were obtained, not known facts in the literature.

Response 14: Revised as required.

Round 2

Reviewer 1 Report

Comments and Suggestions for Authors

Issues previously raised have been addressed sufficiently.

Author Response

Thank you very much for taking the time to review this manuscript.

Reviewer 4 Report

Comments and Suggestions for Authors

Not all my comments were applied. I put my comments here again. They need to be implemented. 

In this paper, the effect of alloying elements on the adsorption and dissociation behaviors of hydrogen molecules on the bcc-Fe (001) surface was investigated using first-principles calculations. It was found that H2 molecule can easily dissociate on the hollow site. The dissociated hydrogen atoms bond with the surrounding metal atoms. Doping Cr and Mo atoms on the surface reduced the H2 molecule adsorption energy, which promotes the H2 molecule adsorption and dissociation. It was also shown that only one or two Ni atoms doping on the surface improves the adsorption energies, which can hinder the H2 molecule adsorption and dissociation. However, three and four Ni atoms doping on the surface are beneficial to the H2 molecule adsorption and dissociation. Recommendations were made to control the nickel content in Ni-Cr-Mo steel to improve the hydrogen embrittlement resistance of the steel.

 The paper could be considered for publication in the journal after the following major revisions:

1-Check the English of the paper.

2--use third person. Avoid using phrases like “We found” etc.

3-A better combination of keywords should be employed, especially avoid using failure analysis.

4-Define in the abstract what parameters were investigated, before briefly mentioning the results of such tests. Like mention that doping was done with how many atoms of what and emphasis on the experimental conditions in terms of process parameters.

5-Introduciton should be strengthened. To modify this section the following documents can be consulted:

-(2022). Water jet impact damage mechanism and dynamic penetration energy absorption of 2A12 aluminum alloy. Vacuum, 206, 111532. doi: https://doi.org/10.1016/j.vacuum.2022.111532

-(2019). A nonlinear attachment-detachment model with adsorption hysteresis for suspension-colloidal transport in porous media. Journal of Hydrology, 578, 124080. doi: https://doi.org/10.1016/j.jhydrol.2019.124080

6-Better define the rationale of the work at the end of the introduction by linking the literature with the undertaken research.

7-it should be “Materials and Computational method”. Introduce the materials first. And make them as two subsections.

8-reference the formula that is used.

9-figure caption 2 is long and tedious.

10-figure 4 shows that Bader charge are rather small by doping. It is even negative for Ni addition. Can the authors elaborate more on this?

11-figure 6 is a bit messy. Make them more visible.

12-consult the following references in the discussion section:

-(2020). Evolution of crystallographic orientation, precipitation, phase transformation and mechanical properties realized by enhancing deposition current for dual-wire arc additive manufactured Ni-rich NiTi alloy. Additive Manufacturing, 34, 101240. doi: https://doi.org/10.1016/j.addma.2020.101240

-(2020). Hydrogen embrittlement behavior of SUS301L-MT stainless steel laser-arc hybrid welded joint localized zones. Corrosion Science, 164, 108337. doi: https://doi.org/10.1016/j.corsci.2019.108337

13-have the authors considered the atomic size difference of different doping? Or only bond energy was considered.

14-conclusions are long and tedious. Make them concise and shorter. It should reflect the results that were obtained, not known facts in the literature.

Comments on the Quality of English Language

English needs to be improved. 

Author Response

Dear Reviewer,

Thank you very much for taking the time to review this manuscript. I have made the revisions to the best of my ability in accordance with your requirements. However, there were a few suggestions that were not consistent with the main focus of the paper, and I did not revise them. Thank you for your understanding.

Comments 1: Check the English of the paper.

Response 1: Checked and corrected as required.

Comments 2: use third person. Avoid using phrases like “We found” etc.

Response 2: Checked and corrected as required.

Comments 3: A better combination of keywords should be employed, especially avoid using failure analysis.

Response 3: Revised as required.

Comments 4: Define in the abstract what parameters were investigated, before briefly mentioning the results of such tests. Like mention that doping was done with how many atoms of what and emphasis on the experimental conditions in terms of process parameters.

Response 4: At present, it is difficult to observe the phenomenon at the atomic level by the experimental methods. We have not yet found the effect of the number of alloying elements on hydrogen adsorption in the experiment. First-principles calculation method is mainly to explain the interaction mechanism between different elements at the atomic level. It can provide a new perspective for researchers to understand hydrogen embrittlement.

Comments 5: Introduction should be strengthened. To modify this section the following documents can be consulted:

-(2022). Water jet impact damage mechanism and dynamic penetration energy absorption of 2A12 aluminum alloy. Vacuum, 206, 111532. doi: https://doi.org/10.1016/j.vacuum.2022.111532

-(2019). A nonlinear attment-detachment model with adsorption hysteresis for suspension-colloidal transport in porous media. Journal of Hydrology, 578, 124080. doi: https://doi.org/10.1016/j.jhydrol.2019.124080

Response 5: Revised as required.

Comments 6: Better define the rationale of the work at the end of the introduction by linking the literature with the undertaken research.

Response 6: Revised as required.

Comments 7: it should be “Materials and Computational method”. Introduce the materials first. And make them as two subsections.

Response 7: Our research object is bcc-Fe (001) surface, which has been introduced in the results and discussions.

Comments 8: reference the formula that is used.

Response 8: Revised as required.

Comments 9: figure caption 2 is long and tedious.

Response 9: Figure 2 shows the three configurations after structural relaxation and differential charge density of H2 on the top, hollow and bridge sites.

Comments 10: figure 4 shows that Bader charge are rather small by doping. It is even negative for Ni addition. Can the authors elaborate more on this?

Response 10: The arrangements of extranuclear electrons are [Ar]3d6 4s2 for Fe, [Ar]3d8 4s2 for Ni, [Ar]3d5 4s1 for Cr, and [Kr]4d4 4s1 for Mo. The Ni atom has more electrons outside the nucleus and is less attractive to the hydrogen atom.

Comments 11: figure 6 is a bit messy. Make them more visible.

Response 11: Revised as required. The density of states diagram is difficult to be clear.

Comments 12: consult the following references in the discussion section:

-(2020). Evolution of crystallographic orientation, precipitation, phase transformation and mechanical properties realized by enhancing deposition current for dual-wire arc additive manufactured Ni-rich NiTi alloy. Additive Manufacturing, 34, 101240. doi: https://doi.org/10.1016/j.addma.2020.101240

-(2020). Hydrogen embrittlement behavior of SUS301L-MT stainless steel laser-arc hybrid welded joint localized zones. Corrosion Science, 164, 108337. doi: https://doi.org/10.1016/j.corsci.2019.108337

Response 12: Revised as required.

Comments 13: have the authors considered the atomic size difference of different doping? Or only bond energy was considered.

Response 13: For the influence of different elements, we only do the single atom doping. The atomic size difference has little effect.

Comments 14: conclusions are long and tedious. Make them concise and shorter. It should reflect the results that were obtained, not known facts in the literature.

Response 14: Revised as required.